# Non-saturating quantum magnetization in Weyl semimetal TaAs

Cheng-Long Zhang[1], C.M. Wang[2,3,4], Zhujun Yuan [1], Xitong Xu[1], Guangqiang Wang[1], Chi-Cheng Lee[5,6], Li Pi[7], Changying Xi[7], Hsin Lin[5,6], Neil Harrison[8], Hai-Zhou Lu[2,3,9], Jinglei Zhang[7] & Shuang Jia[1,10,11]

Detecting the spectroscopic signatures of relativistic quasiparticles in emergent topological materials is crucial for searching their potential applications. Magnetometry is a powerful tool for fathoming electrons in solids, by which a clear method for discerning relativistic quasiparticles has not yet been established. Adopting the probes of magnetic torque and parallel magnetization for the archetype Weyl semimetal TaAs in strong magnetic field, we observed a quasi-linear field dependent effective transverse magnetization and a non-saturating parallel magnetization when the system enters the quantum limit. Distinct from the saturating magnetic responses for non-relativistic quasiparticles, the non-saturating signals of TaAs in strong field is consistent with our newly developed magnetization calculation for a Weyl fermion system in an arbitrary angle. Our results establish a high-field thermodynamic method for detecting the magnetic response of relativistic quasiparticles in topological materials.

[1] International Center for Quantum Materials, School of Physics, Peking University, 100871 Beijing, China. [2] Shenzhen Institute for Quantum Science and Engineering and Department of Physics, Southern University of Science and Technology, 518055 Shenzhen, China. [3] Shenzhen Key Laboratory of Quantum Science and Engineering, 518055 Shenzhen, China. [4] Department of Physics, Shanghai Normal University, 200234 Shanghai, China. [5] Centre for Advanced 2D Materials and Graphene Research Centre, National University of Singapore, 6 Science Drive 2 117546, Singapore. [6] Department of Physics, National University of Singapore, 2 Science Drive 3 117542, Singapore. [7] Anhui Province Key Laboratory of Condensed Matter Physics at Extreme Conditions, High Magnetic Field Laboratory of the Chinese Academy of Sciences, Hefei 230031, China. [8] National High Magnetic Field Laboratory, Los Alamos National Laboratory, MS E536, Los Alamos, NM 87545, USA. [9] Center for Quantum Computing, Pengcheng Laboratory, 518055 Shenzhen, China. [10] Collaborative Innovation Center of Quantum Matter, 100871 Beijing, China. [11] CAS Center for Excellence in Topological Quantum Computation, University of Chinese Academy of Sciences, 100190 Beijing, China. These authors contributed equally: Cheng-Long Zhang, C. M. Wang Correspondence and requests for materials should be addressed to H.-Z.L. (email: luhz@sustc.edu.cn) or to J.Z. (email: zhangjinglei@hmfl.ac.cn) or to S.J. (email: gwljiashuang@pku.edu.cn)

The low-energy states of electrons in topological materials can be described as a series of quasiparticles, which obey different representations of the Dirac equation[1–5]. One kind of the three-dimensional (3D) massless quasiparticle is Weyl fermion, which has been discovered in topological Weyl and Dirac semimetals[6–13]. The Weyl quasiparticles occur in the vicinity of a finite number of band touching points, dubbed Weyl nodes, in these topological semimetals. The unique topological nature of the Weyl semimetal promises many novel properties belonging to the massless quasiparticles, such as linear energy dispersion, monopoles, and Fermi arcs on the surface[14–16]. Of particular, the 0th Landau bands (LBs) of the Weyl fermions in strong magnetic field are purely chiral modes[17–19]. These one-dimensional (1D), chiral LBs are expected to exhibit a negative longitudinal magnetoresistance (MR) as a signature of the long-sought chiral anomaly in quantum field theory[20]. Realizing the chiral anomaly in solids has inspired intensive experimental activities on topological semimetals in strong magnetic field[21–27]. Nevertheless, the MR in the quantum limit (QL) also depends sophisticatedly on the nature of the impurity scattering[28] and thus it cannot give the information of the quasiparticles' spectrum deterministically[29–31]. Actually the alleged linear positive longitudinal magneto-conductance in theory has not been observed experimentally in the QL of topological semimetals.

By contrast, the magnetic responses of the electrons are much less complex, because they do not interplay with impurity scattering. Indeed they are simply determined by the derivatives of the electrons' thermodynamical potential $\Omega$ with respect to magnetic field $H$, and they have been used to probe the properties of the Fermi surface, including the relativistic aspects[32–34]. Previous studies claimed that the non-trivial Berry phase accounts for the non-zero extrapolation of the quantum oscillations (QOs) in topological semimetals[28,35–39]. This Landau-fan diagram-based criterion is often blurred by complicated resistivity tensors[28]. By contrast, we suggest a method for detecting relativistic quasiparticles by using of the magnetic responses in this paper. We focus on the magnetic response of the 0th LB in a sufficiently strong magnetic field when all the rest LBs have left the Fermi surface. In such extreme condition, the crossing bands show sheerly different magnetic response than that of trivial parabolic bands. To illustrate the difference, we choose TaAs as a prototype topological semimetal hosting well-defined Weyl quasi-electrons. The cross section area of the Weyl pocket is sufficiently small so that a steady magnetic field can approach the QL within a large deviation of the angle in a magnetic torque measurement. We show that the effective transverse ($M_T$) and parallel magnetizations ($M_{||}$) of TaAs are quasi-linear field dependent beyond the QL. Consistent with our calculations, these non-saturating magnetic responses are distinct from that for non-relativistic quasiparticles. Our observation serves as a thermodynamic method for discerning the relativistic quasiparticles in emergent topological materials.

## Results

### Theory of high-field magnetization.
Before discussing the experimental results, we first compare the pictures of the magnetic response for the relativistic and non-relativistic electron systems (Supplementary Note 2). Figure 1a, b is the sketches of the bands for the non-relativistic electrons and holes in a magnetic field, in which the magnetization is contributed by both conduction and valence bands, which are separated by a band gap. To calculate the full-response magnetization, we include the contributions of the valence bands and conduction bands. When the field increases, the LBs successively leave the Fermi surface (Fig. 1b), leading to the oscillatory $M_{||}$ around zero and the

dropping $M_T$ with increasing field (Fig. 1e, f) until all of the LBs have left the Fermi surface except for the lowest one. The Fermi energy $E_F$ almost remains intact in low fields when there are a lot of LBs below it, so the low-field simulations for a fixed $E_F$ and for a fixed carrier concentration ($N_c$) are almost identical. However, the change of $E_F$ in strong magnetic fields cannot be ignored when the system enters the QL (Supplementary Note 2). Here we consider two possible constraints for the calculations: imposing the conservation of $N_c$ or fixing $E_F$. If $E_F$ is fixed, the zero-point energy will lift the 0th LB above $E_F$ beyond a critical magnetic field, leading to vanishing $M_{||}$ and $M_T$ (red dot lines in Fig. 1e, f). If $N_c$ is fixed, $E_F$ will be pinned at the edge of the 0th LB regardless of the increase of the magnetic field. Under this constraint, $M_{||}$ will saturate and $M_T$ will be a constant (blue lines in Fig. 1e, f). This high-field behavior is obvious in Supplementary Eqs. (10) and (62), where both the contributions from valance and conduction bands saturate at high fields. In other words, both $M_{||}$ and $M_T$ for the non-relativistic electrons are invariant in the QL due to non-relativistic band dispersion. This picture is tenable for an artificial case where two parabolic bands touch at a point, as shown in Fig. 1c, d. Our calculations prove that the magnetization behaviors of this case in the QL is similar to that with a gap (Supplementary Note 2). The profile of the saturating $M_{||}$ and $M_T$ in the QL has been observed in 3D massive (gapped) bulk systems, such as Bi, sulfur-doped $Bi_2Te_3$, and InSb[34,40,41].

By contrast, gapless Weyl fermions contribute to the magnetization in a different way due to the band inversion and relativistic spectrum (Fig. 1g, h). Both $M_{||}$ and $M_T$ oscillate around a mean value in low magnetic fields (Fig. 1k, l). When all the N > 0 LBs have left $E_F$, both chiral modes contribute to the magnetic response (Fig. 1h), leading to non-saturating $M_{||}$ and $M_T$ in strong magnetic fields. We then consider a generalized case in which the Dirac model with a mass term is adopted, instead of the massless Weyl model (Fig. 1i, j). We are interested in whether this behavior still exists for a generalized model, namely the Dirac model with a mass term. The simulations for that model show a similar and consistent result compared with that of the Weyl case only when the gap is much smaller than the cutoff energy ($\Lambda$). In the large-gap limit, it is natural to expect that the model reduces to a trivial case as shown in Fig. 1a. This means that the relativistic feature, e.g., the linearity of the energy dispersion, should present in the Dirac model and thereupon the non-saturating magnetization in the QL is expected. The essential difference of the magnetic responses in the QL stems from the nature of relativistic electrons. In the following theoretical part, the origin of this difference will be discussed in detail, by including both the conduction and valence Weyl bands.

### Magnetic torque and effective transverse magnetization.
The depiction above sheds light for understanding the magnetic torque signals of TaAs. The magnetic torque $\tau$ is defined as $-\frac{\partial \Omega}{\partial \theta}$, where $\theta$ is the angle of the magnetic field with respect to the $c$-axis. In a macroscopic expression $\tau$ is formulated as a mechanic torsional torque, $\tau = V\mathbf{M} \times \mu_0\mathbf{H} = \mu_0 V H M_T$. Figure 2a, b shows $\tau$ and $M_T$ against magnetic field at several representative tilted angles, respectively. Clear de Haas–van Alphen (dHvA) QOs superpose on a large diamagnetic background at different temperatures. The dHvA oscillations are predicted by Landau's theory for diamagnetism, and this Landau diamagnetism is small in metals and usually smeared by other magnetic signals due to the spin contribution (Pauli paramagnetism) and the core shell (Lamor diamagnetism). However, the electron mass in semimetals is renormalized by the Fermi liquid theory and the orbital contribution of the Landau diamagnetism gains a huge enhancement entailing a square relation[42]. In TaAs, the cyclotron

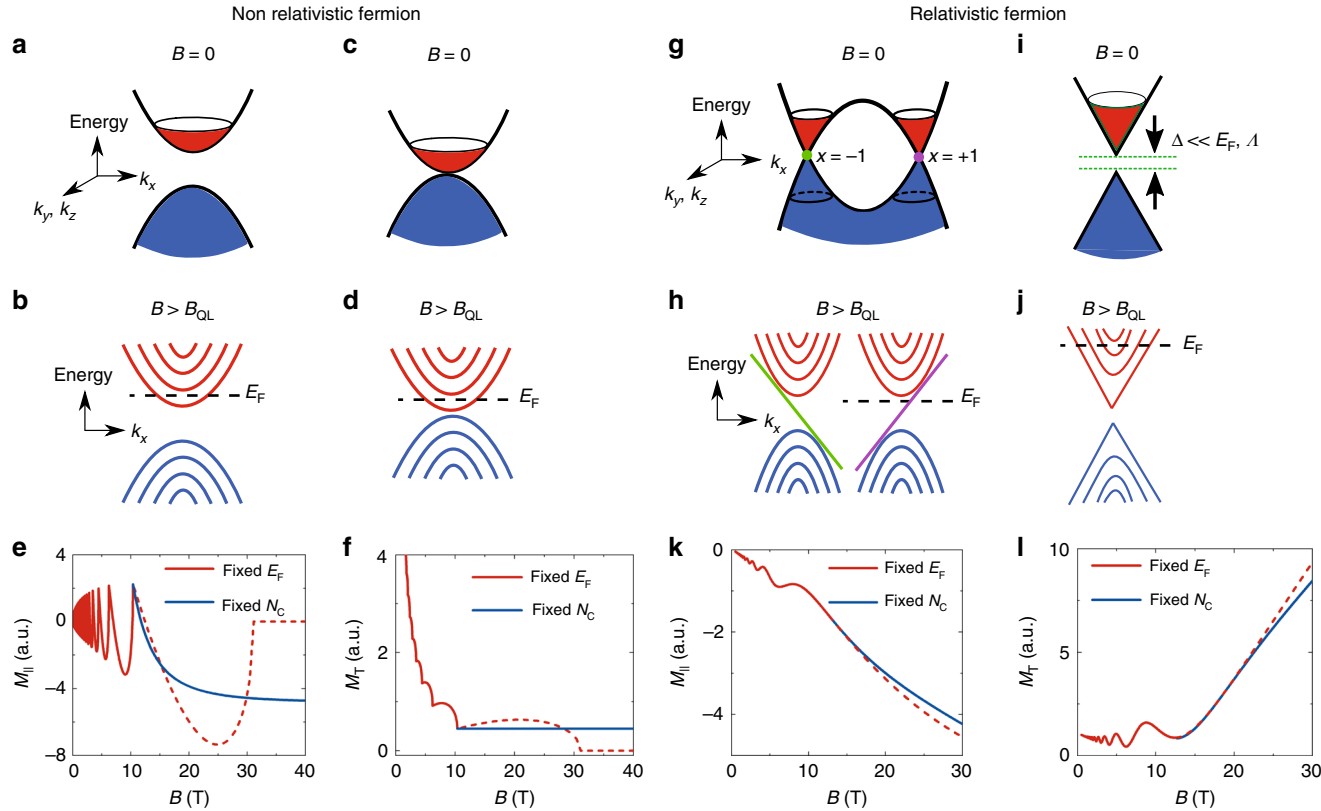

**Fig. 1** Magnetic responses of the non-relativistic and relativistic fermions. **a** The energy bands of non-relativistic (parabolic-band) fermions in zero magnetic field. **b** The LBs of non-relativistic fermions form in a magnetic field. Only the 0th LB crosses the $E_F$ in the QL. **c** Two energy bands of non-relativistic (parabolic-band) fermions touch each other in a momentum point in zero magnetic field. **d** The LBs of non-relativistic fermions touch each other in a magnetic field. **e, f** Calculated parallel magnetization ($M_\parallel$) and effective transverse magnetization ($M_T$) of non-relativistic fermions (the two cases show identical behaviors in the QL, here we just show a representative one for brevity) as functions of magnetic field. We used two constraints for the calculation in the QL: fixed the $E_F$ (red line) and fixed carrier density ($N_c$) (blue line). **g** A typical schematic energy band of a pair of type I Weyl nodes in zero magnetic field. **h** A series of LBs for a pair of Weyl nodes form in a magnetic field. Two 0th LBs are entirely chiral (green and violet). **i** A typical relativistic fermion band (Dirac model) with a small gap. The gap is much smaller than the cutoff energy $\Lambda$. This model also gives out the same non-saturating behavior as we see in the Weyl case. **j** The corresponding series of LBs for a simple Dirac model with a small gap in a magnetic field. **k, l** Calculated $M_\parallel$ and $M_T$ as functions of magnetic field, respectively. The two cases show identical behaviors in the QL, and here we just show a representative one for brevity

mass $m^*$ is 0.06 $m$, where $m$ is the free electron mass (Supplementary Note 3), and therefore the enhancement takes the form of $\frac{\chi_{Landau}}{\chi_{Pauli}} \sim (m/m^*)^2$ [42] leading to a value about 300. Bearing in mind that the Lamor diamagnetization and Pauli paramagnetization have the same order, we believe that the magnetization in TaAs is dominated by the Landau diamagnetism in the whole range of magnetic field. In essence, only one frequency is detected at small angles and no dHvA QOs are found beyond the QL of these QOs. Instead, $M_T$ linearly increases with respect to $H$, with a pronounced slope change after entering the QL. The change of the dHvA QOs and the magnetic response at different temperatures for a tilted angle $\theta = 34.5°$ are shown in Fig. 2c, d, respectively. The dHvA QOs decay at higher temperatures while $\tau$ and $M_T$ remain intact beyond the QL. It is noteworthy that the extrapolated intercept of the linear $M_T$ is far less than zero, which indicates that this featureless $M_T$ in strong magnetic fields is not the same as the commonly observed Lamor diamagnetization proportional to low fields. Consistent with the band structure calculations, our data indicate that the extremal cross sections of the Fermi pockets enclosing the W1 and W2 nodes are 9.5 T and 3.3 T, much smaller than that of NbAs[38]. The lower Fermi level in TaAs makes the Weyl physics dominate (Supplementary Note 3),

namely the chirality and the linearity of the bands are well defined. The well-defined Weyl bands are essential in our calculations for the non-saturating magnetization in the QL. We think that both the W1 and W2 pockets contribute to the observed non-saturating behavior in the QL (Supplementary Note 2).

**Differential susceptibility**. To understand the non-saturating $M_T$, we plot the differential susceptibility $\chi_T = \partial M_T / \partial H$ at different angles and temperatures with respect to the field in Fig. 3a, b, respectively. With increasing field, the $\chi_T$ evolves from QOs to a plateau when the QL is achieved at different angles. The height of the plateaus remains intact with increasing temperatures, which is sheerly different from the damping amplitude of the dHvA QOs in lower fields (Fig. 3b). The plateau and the slope show a good linearity of the transverse magnetization in the QL as shown in Fig. 3b. Figure 3c shows that the heights of the plateaus above the QL at different angles can be well fitted by a relation of $\sin 2\theta$ for $\theta < 60°$, consistent with the expression of $\tau = V\mathbf{M} \times \mu_0 \mathbf{H}$, which can be reformulated as $\frac{1}{2}\Delta\chi\mu_0 H^2 V \sin 2\theta$ ($\Delta\chi = \chi_\parallel - \chi_\perp$).

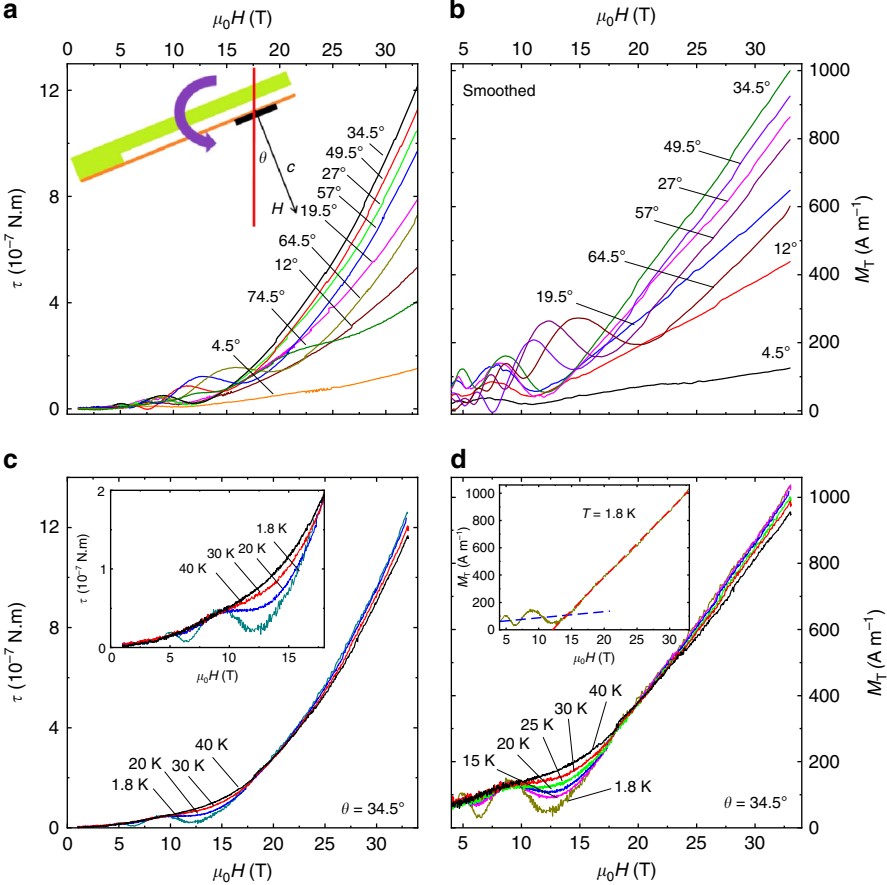

**Fig. 2** $\tau$ and $M_T$ versus magnetic field at different temperatures and angles. **a, b** $\tau$ and $M_T$ at 1.8 K at different tilted angles, respectively. Inset in **a** shows the rotation setup where the angle $\theta$ is defined as the tilt off the $c$-axis. The curves in **b** have been smoothed. **c, d** $\tau$ and $M_T$ in a fixed angle ($\theta = 34.5°$) at different temperatures, respectively. Inset in **c** shows a zoom-in in low fields where strong temperature dependent dHvA QOs superpose on a parabolic background. Inset in **d** shows the $M_T$ curve at 1.8 K, where two dashed coloured lines show the low- and high-field slopes. The slope takes an obvious enhancement near the QL

**Simulations based on a Weyl Hamiltonian.** Now we show that the magnetic signals of TaAs dovetail our calculations for the 3D Weyl fermions in the strong-field limit. Previous calculations only addressed $M_\parallel$ over the full range of magnetic field for different types of band contacting[37,43,44], while here we formulate a more general magnetization theory for $M_\parallel$, $M_T$, and $\tau$ in an arbitrary angle for Weyl semimetals in which the band structure is also taken into account. The full version of the theory including a version for multiple Weyl nodes is presented in Supplementary Note 2 and here we give the conclusion in the strong-field limit. The Hamiltonian for a single node of 3D massless Weyl fermions can be formulated as

$$\mathcal{H} = v_a p_x \sigma_x + v_b p_y \sigma_y + v_c p_z \sigma_z, \tag{1}$$

where $v_{a,b,c}$ take into account the anisotropy of the Fermi velocity, $p_{x,y,z}$ are the momentum, and $\sigma_{x,y,z}$ are the Pauli matrices. Previous studies mainly focused on the magnetization at low fields (the magnetization will be quenched at high fields where the system enters the QL), which includes the Berry paramagnetism for the Weyl semimetals as pointed out in ref. [38]. The competition between the Berry paramagnetism and other magnetic signals (Landau diamagnetism and Lamor diamagnetism) may lead to positive or negative torque signals at low fields. Moreover, the recent theoretical calculations on a lattice model revealed that both the magnitude and sign of the orbital contribution of magnetization depend on the exact value of the Fermi energy in a

Weyl/Dirac semimetal[45]. These theoretical analyses are consistent with our measurements for other members of the TaAs family (Supplementary Note 4). On the other hand, the profile of the magnetization in the QL can be more precisely described, and we find the magnetization of this model in the QL (Supplementary Note 2) as

$$M_\parallel \propto -B \ln \Gamma \tag{2}$$

for the parallel component, and

$$M_T \propto -B \ln \Gamma \frac{\partial \Delta}{\partial \phi} \tag{3}$$

for the transverse component. Here $\Gamma = 2\Lambda \ell_B / v_b \sqrt{2\Delta}$, where $-\Lambda$ is the cutoff energy of the valence band, $\Delta = (v_a/v_b) \cos \alpha \cos \theta + (v_c/v_b) \sin \alpha \sin \theta$, where $\alpha = \tan^{-1}\left(\frac{v_c}{v_a} \tan \theta\right)$ and $\ell_B = \sqrt{\hbar/eB}$ is the magnetic length. We can clearly see that the non-saturating (Supplementary Eqs. (34) and (35)) magnetization comes from the lower Weyl cone, which is treated as negative energy band in the QL when the magnetization of the higher Weyl cone is quenched[38]. We denote that the magnetization that comes from the lower Weyl cone is compatible with the Landau theory with negative-energy LBs. Therefore, we conclude that the behavior of magnetization in the QL does not depend on detailed material parameters, and thus it is a universal spectral fingerprint.

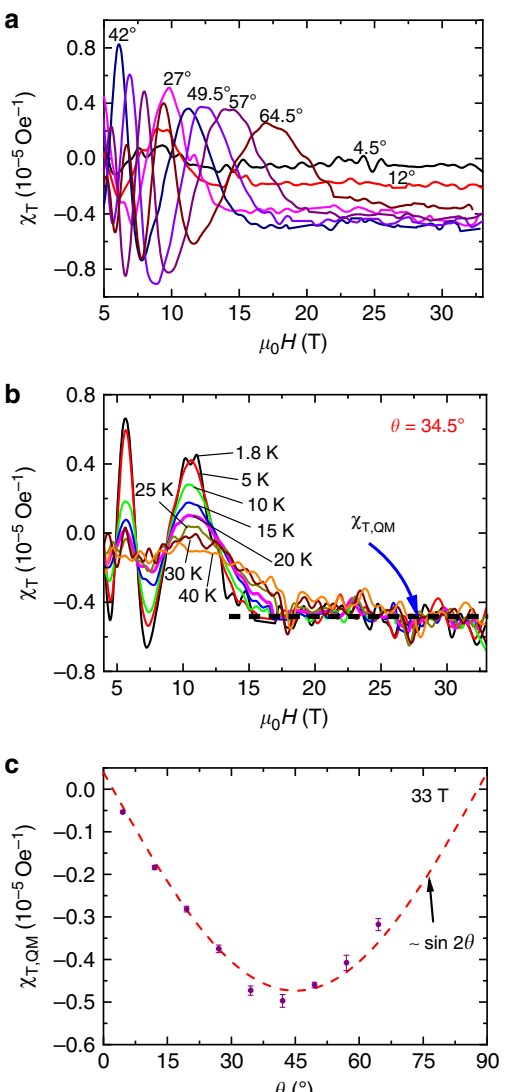

**Fig. 3** Differential effective susceptibility ($\chi_T$) for TaAs. **a** $\chi_T$ at different angles. **b** $\chi_T$ for $\theta = 34.5°$ at different temperatures. **c** The heights of the $\chi_T$ plateaus at 33 T versus angles. Red dashed line shows the fitting with a relation of $\sin 2\theta$. The error bars were obtained from the standard deviations of intercepts when fitting to the step behaviors in the quantum limit

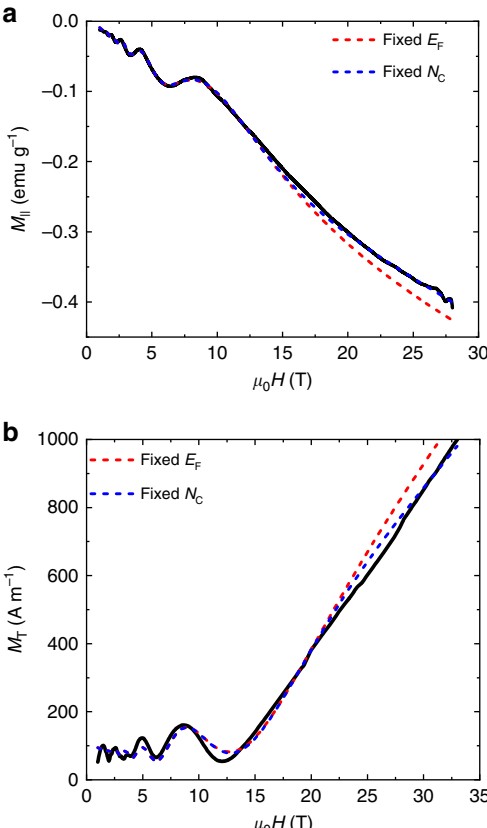

**Fig. 4** Comparison between experiments and theory for the $M_{||}$ and $M_T$ of TaAs. **a** $M_{||}$ (black line) and **b** $M_T$ (black line). The blue dashed lines represent the theoretical results with the constraint of the fixed $E_F$ in the full range of magnetic field, while the red dashed lines represent the results with the constraint of the fixed $N_c$ in strong fields

Then we check whether our calculation can well fit the experimental results for TaAs. Figure 4a shows that the measured $M_{||}$ for H||c on another sample in strong pulsed magnetic field. The data can be well fitted by using Eq. (2) where we assume a fixed $N_c$ (Supplementary Note 6). Our theoretical simulation of the formulas for the 3D Weyl fermions reproduces the dHvA QOs at low fields and a significantly enhanced $M_{||}$ in the QL. For comparison, the $M_{||}$ for the 3D non-relativistic electrons with parabolic energy dispersion [see Supplementary Eq. (10)] saturates in the QL. On the other hand, we can well fit $M_T$ in Fig. 4b by Eq. (3) (Supplementary Note 6). We emphasize that the fittings with a fixed $N_c$ or $E_F$ give a similar trend in strong fields, which is distinct from the saturating magnetization of non-relativistic electrons.

## Discussion

The linear non-saturating magnetization in the QL is of particular interest because it is beyond Landau's theory of magnetization for classical electrons[46] (see Supplementary Note 2 for a fundamental theoretical approach for the classical theory of magnetization). Such feature has never been observed in semimetals with over-lapped parabolic electron and hole bulk bands in strong magnetic fields[41,47–50]. Our calculations and experiments demonstrate the unique magnetic response in the QL for the linear energy dispersive of the 0th LB of Weyl fermions. Recently emergent topological materials with relativistic quasiparticles for potential applications in general manifest small Fermi surfaces in which the QL can be accessed in a constant magnetic field. Their relativistic nature of quasiparticles is difficult to identify by spectroscopic and electrical transport techniques. The magnetization may serve as a method for detecting relativistic quasiparticles in emergent materials. Finally, we emphasize that our calculation and explanation are based on a single particle scenario. How our theory of one-particle physics can be applied to many-body effects, such as to excitons[55], will be a topic of great interest.

## Methods

**Sample preparation and characterization**. TaAs has 12 pairs of Weyl nodes, which are divided into four pairs of W1 and eight pairs of W2. The $E_F$ of the as-grown single-crystalline TaAs is close to the two types of Weyl nodes, which are separated 13 meV in energy space, therefore its Fermi surface consists pairs of Weyl electron pockets (Supplementary Fig. 8)[51,52]. We prepared the single crystals of TaAs by the standard chemical vapor transfer (CVT)[53,54] in this study. The large single crystals we used for the magnetic torque is shown as an inset in Supplementary Fig. 1. Its polished surface shows the (001) plane, which was confirmed by X-ray diffraction measurements. The single crystal for parallel magnetization

measurements in the pulsed field is as large as $1 \times 1 \times 5$ mm for acquiring the data with higher resolution.

**Measurement of magnetic torque**. The magnetic torque measurements were performed using capacitive cantilever in water-cooled magnet with the steady fields up to 33 T in the Chinese High Magnetic Field Laboratory (CHMFL), Hefei. In order to estimate the background signal from the cantilever and the cable, the empty cantilever was calibrated on the same conditions. The details of the calibration and measurements are shown in Supplementary Note 1.

## Data Availability
The data that support the plots within this paper and other findings of this study are available from the corresponding author upon reasonable request.

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

## Acknowledgements
C-L.Z. appreciates Lu Li's crucial comments on small angle torque theory and treats his research work as precious homework from Shui-Fen Fan. J.Z. thanks Dr. Lin Jiao for sharing his analysis program with us. C.-M.W. thanks lots of discussions from Yuriy Sharlai. S.J. is supported by the National Natural Science Foundation of China No. U1832214, No.11774007, the National Key R&D Program of China (2018YFA0305601) and the strategic Priority Research Program of Chinese Academy of Sciences, Grant No. XDB28000000. J.Z. was supported by the Youth Innovation Promotion Association CAS (grant number 2018486) the Scientific Instrument Developing Project of the Chinese Academy of Sciences (Grant No.YJKYYQ20180059) and the Innovative Program of Development Foundation of Hefei Center for Physical Science and Technology (2017FXCX001). H.Z.L was supported by the Guangdong Innovative and Entrepreneurial Research Team Program (Grant No. 2016ZT06D348), National Key R & D Program (Grant No. 2016YFA0301700), the National Natural Science Foundation of China (Grants No. 11574127), and the Science, Technology and Innovation Commission

of Shenzhen Municipality (Grant No. ZDSYS20170303165926217, No. JCYJ20170412152620376). C-M.W. is supported by the National Natural Science Foundation of China (Grant No. 11474005). H.L. acknowledges the Singapore National Research Foundation for the support under NRF Award No. NRF-NRFF2013-03. The National Magnet Laboratory is supported by the National Science Foundation Cooperative Agreement no. DMR-1157490, the State of Florida, and the US Department of Energy. Work at Los Alamos National Laboratory was supported by the Department of Energy, Office of Science, Basic Energy Sciences program LABLF100 "Science of 100 Tesla." C-L.Z and C-M.W. contributed equally to this work.

## Author contributions

C-L.Z. and S.J. conceived the project. J.Z. performed the magnetic torque experiment with help from C-L.Z., L.P., C.X., and S.J.; C-L.Z. did the electrical transport measurements. X.X. and G.W. performed the low-field magnetization measurements. N.H. performed the ac parallel magnetization in pulsed field. Z.Y. grew the single-crystal samples; C-C.L. and H.L. performed first principles band structure calculations; C-L.Z. wrote the classical high-field magnetization theory; C.M.W. and H-Z.L. wrote the full magnetization theory and did all theoretical simulations; C-L.Z., C.M.W., J.Z., H-Z.L., and S.J. wrote the paper. S.J. was responsible for the overall direction, planning, and integration among different research units.

## Additional information

**Competing interests:** The authors declare no competing interests.

