## [Peer Review File · Nature Communications]

Reviewers' comments:

Reviewer #1 (Remarks to the Author):

This manuscript reports magnetization measurements in a Weyl semimetal TaAs. A nonsaturating linear field dependence of the magnetization is observed and the authors interpret this as evidence of the chiral lowest Landau level (LLL) and thus Weyl band touching points. I find this work to be worthwhile and interesting. In principle, this may be published in Nature Communications. However, improvements are necessary before publication, in my opinion.

1. The paper is very poorly written. There are multiple typos and multiple sentences that barely make sense to me. The writing needs to be cleaned up significantly before the paper can be published.

2. The authors base their analysis on a simple model Weyl Hamiltonian. TaAs electronic structure is significantly more complex than that, there are multiple Weyl node pairs at different energies. At a minimum, the authors should include estimates, based on published TaAs electronic structure, of the magnetic field strength necessary to enter the quantum limit (i.e. only the LLL at the Fermi energy).

Reviewer #2 (Remarks to the Author):

The authors performed magnetic torque and magnetization measurements on the Weyl semimetal TaAs up to 33 T. A theory calculation to explain the data was also provided and the distinction between massive and Weyl electrons was discussed. The magnetization data and the theory calculation matched well, as this is a non-interaction system and TaAs is a well-established Weyl semimetal. The paper is well presented and nicely written in overall.

However, I can not support this work to be published in Nat. Comm as this is not the first work to distinguish massive and massless electrons in Weyl semimetals in magnetic torque measurement. Specifically, the authors claimed that the use of magnetometry method to discern massless Weyl fermion in the quantum limit was the first time, which is not a correct statement. Ref. 38 (Moll et al Nat. Comm. 7, 12492 (2016)) measured torque up to 65 T and they calculated the difference between massive, Dirac and Weyl electronics to distinguish them. Even though Ref. 38 did not measure magnetization directly, I didn't see that new Physics or exotic phenomenon was revealed in the current work above the findings in Ref. 38. As the fact that TaAs is a Weyl semimetal is well confirmed by various probes, the matching between theory and experiments is not very surprising. I also have other specific comments as following.

1). The chemical potential of the sample is almost at the Lifshiz energy near W1 Weyl point, so is W1 not contributing to the non-saturation of the magnetization? About analyzing the dHvA data, I hope the authors can provide more details about the carrier density for each pocket and and discuss whether they contribute to the torque data in the quantum limit.

2). For the torque data below the quantum limit, Why is the trend bending up (positive) for TaAs as this is a Weyl semimetal?

3). For the transverse magnetic susceptibility, the author gave a quantum value in Fig. 3B, is there any significance about the number? It was not discussed in the paper. Also, at zero field, is the transverse magnetic susceptibility also very weakly dependent on temperature from 1.5 K to 40 K. I suspect that the temperature dependence is also weak as this is a paramagnetic metal.

Reviewer #3 (Remarks to the Author):

The main result of this paper is the observation of a non-saturating torque magnetization at high magnetic field in TaAs, a semimetal known to present Weyl nodes in its band dispersion. The authors conclude that they have found a new signature of 3D massless Weyl fermions in the quantum limit, namely non-saturating linear magnetization.

My main objection to this conclusion is that it ignores what is experimentally known about other semimetals, which do NOT host Weyl nodes. The most obvious case is graphite. It is a semimetal with a topologically trivial band structure. When pushed to the quantum limit, it undergoes a field-induced phase transition documented extensively during the last three decades. There is no Weyl fermion in graphite. Yet, its magnetization is featureless, quasi-linear and non-saturating up to 30 T. (See Fig. 4 in Uji et al., *Physica B* 246-247, 299 (1998)). This looks like very similar to what the present authors find in the case of TaAs.

Two lessons are to be drawn.

First of all, even in presence of topologically-trivial electrons, one can observe a non-saturating magnetization.

Second, diamagnetism in semimetals (including TaAs) is caused by orbital currents, which may completely overshadow the contribution of Landau-level electrons, the principal player in the theoretical scenario put forward in this paper. How to explain otherwise the absence of any signature in magnetization by an electronic reorganization such an excitonic phase transition (*Sci. Rep.* 7: 1733 (2017)) in graphite?

Reply to Reviewer #1

[Remark 1-1]: *This manuscript reports magnetization measurements in a Weyl semimetal TaAs. A nonsaturating linear field dependence of the magnetization is observed and the authors interpret this as evidence of the chiral lowest Landau level (LLL) and thus Weyl band touching points. I find this work to be worthwhile and interesting. In principle, this may be published in Nature Communications. However, improvements are necessary before publication, in my opinion.*

[Reply 1-1]: We are excited to see that the reviewer acknowledges that "I find this work to be worthwhile and interesting". We thank the reviewer for his/her recommendation for publication.

[Remark 1-2]: *The paper is very poorly written. There are multiple typos and multiple sentences that barely make sense to me. The writing needs to be cleaned up significantly before the paper can be published.*

[Reply 1-2]: In this revised version, we have carefully polished our paper. The typos are corrected, and questionable sentences are rewritten.

[Remark 1-3]: *The authors base their analysis on a simple model Weyl Hamiltonian. TaAs electronic structure is significantly more complex than that, there are multiple Weyl node pairs at different energies. At a minimum, the authors should include estimates, based on published TaAs electronic structure, of the magnetic field strength necessary to enter the quantum limit (i.e. only the LLL at the Fermi energy).*

[Reply 1-3]: We thank the reviewer for pointing out this underlying multi-band physics in the Weyl semimetal TaAs. The estimated fields for the quantum limit are 9.5 T for W1 and 3.3 T for W2, respectively, based on the dHvA data. Therefore a field higher than 10 T is sufficient above the QL for both pockets. Second, we have recalculated the magnetization based on the realistic model of the Weyl semimetal TaAs which includes the two different types of Weyl nodes (in Fig. S7, now Fig. S8 of this revised version). We found that the total magnetization and torque is the summation of the contributions of individual Weyl nodes. Therefore, the magnetization or torque is still non-saturating in the quantum limit. The theoretical analysis is also applicable for the multi-Weyl-node case. We have added this calculation in the SI (See section II-E of this revised version). The Fermi level is placed at 27 meV above W1, and the energy separation between the two kinds of Weyl points is 13 meV (see section III in the revised version of SI). The calculated results of the cross-sectional areas are listed in Table S1, and the corresponding quantum limits are consistent with the values obtained from the dHvA data.

Reply to Reviewer #2

[Remark 2-1]: *The authors performed magnetic torque and magnetization measurements on the Weyl semimetal TaAs up to 33 T. A theory calculation to explain the data was also provided and the distinction between massive and Weyl electrons was discussed. The magnetization data and the theory calculation matched well, as this is a non-interaction system and TaAs is a well-established Weyl semimetal. The paper is well presented and nicely written in overall.*

[Reply 2-1]: We thank the reviewer for his/her precious time and positive comments.

[Remark 2-2]: *However, I can not support this work to be published in Nat. Comm as this is not the first work to distinguish massive and massless electrons in Weyl semimetals in magnetic torque measurement. Specifically, the authors claimed that the use of magnetometry method to discern massless Weyl fermion in the quantum limit was the first time, which is not a correct statement. Ref. 38 (Moll et al Nat. Comm. 7, 12492 (2016)) measured torque up to 65 T and they calculated the difference between massive, Dirac and Weyl electronics to distinguish them. Even though Ref. 38 did not measure magnetization directly, I didn't see that new Physics or exotic phenomenon was revealed in the current work above the findings in Ref. 38. As the fact that TaAs is a Weyl semimetal is well confirmed by various probes, the matching between theory and experiments is not very surprising.*

[Reply 2-2]: We agree with the reviewer that our work and Ref. [38] share the same goal for detecting Weyl fermions in torque measurements. Our work is partially inspired by the wonderful work in Ref. [38].

However, we find the criterion claimed in Ref. [38] does not apply to other Weyl materials, such as TaAs, NbP, and TaP. Therefore, we come up with the new criterion supported by new measurements and calculations. Specifically, the criterion of distinguishing Weyl fermions in Ref. [38], according to Abstract, is “*The torque changes sign in the quantum limit, signalling a reversal of the magnetic anisotropy that can be directly attributed to the topological nature of the Weyl electrons*”. This signature is shown in the left panel of Fig. R1 below (adapted from Fig. 1a of Ref. [38]). Our measurement in the right panel of Fig. R1 also confirmed Ref. [38]’s sign change of the torque in NbAs.

[Redacted]

Figure R1. The torque data of NbAs adapted from Fig. 1a of Ref. [38] (left) and our data for the torque of NbAs (right). The green circle shows a bending from negative background (as indicated by the green line in the right panel) to positive tendency, which was thought to be a signature for Weyl fermions in Ref. [38].

However, we find that our torque data for other Weyl semimetals does not show the sign change, as shown in Fig. R2 below, for TaAs, NbP, and TaP.

Figure R2. Our data for the torque of TaAs, NbP, and TaP, the background (the green lines) below the quantum limit are positive, so there is no sign change, in contrast with that in NbAs.

These experimental observations against the claim in Ref. [38] drove us to find a more universal criterion in torque measurements for detecting Weyl fermions.

Moreover, there are points that distinguish our work from Ref. [38], which we clarify as follows.

1. We mainly focus on the magnetization in the quantum limit ($B > B_c$, where B_c is the critical field above which the system is in the quantum limit). We find that the non-saturating magnetization is a more universal criterion, as supported by our experimental observations and theoretical calculations.

Ref. [38] focuses on the torque, which is related to the effective transverse magnetization via

$$\tau = BM_T.$$

Hence, even for constant (saturating) M_T , the torque linearly depends on the

magnetic field and shows non-saturating feature. More importantly, the underlying physics of the torque and the non-saturating magnetization are completely different. In this revised version, we have added this discussion of the difference between torque and magnetization.

2. The physics in Ref. [38] focuses on below the quantum limit ($B < B_c$), where the magnetization bends to negative (the so-called Berry para-magnetism). The argument there is: This negative trend arises because the Fermi level is pinned by the magnetic field to the $E=0$ point of the lowest Landau band (in contrast to the parabolic band where the lowest Landau band has the zero-point energy). When entering the quantum limit (from $B < B_c$ to $B > B_c$), the lowest Landau band contributes a huge diamagnetization signal and causes the signal reversal.

We have done a similar simulation for $B < B_c$, even though this regime is not our focus. We used two constraints: one is to fix the Fermi level and the other is to fix the carrier concentration (the latter is the same constraint in Moll et al's paper to obtain the Berry paramagnetism near B_c). We found that it is hard to distinguish the two curves (Fermi level fixed and concentration fixed) for $B < B_c$ because: (1) when crossing the higher Landau bands, the Fermi level is not sensitive to the magnetic field, so in the calculations the two constraints give almost the same result. (2) The actual amplitude of the Fermi level bending down in the field is sensitively dependent on the actual Landau band structure. This may give an intuitive picture why the negative bending (claimed in Ref. [38]) is not a universal behavior in different materials.

We have refined our claim based on the newly added experimental data and further refined theoretical calculations considering more realistic band structures. As we shown in Fig. 1 in this revised version, we have presented various possibilities of band arrangements and calculated the corresponding magnetization responses. Based on all possible cases, the claim is better to be refined as **“the non-saturating magnetization in the quantum limit provides a criterion for discerning relativistic fermions (Weyl case or Dirac case with a small gap) and nonrelativistic fermions (parabolic bands with a gap and two parabolic bands touching at a point)”**.

We will appreciate if the reviewer acknowledges the advance of our work. We believe that with the revisions, our paper deserves to be published in Nature Communications.

[Remark 2-3]: *The chemical potential of the sample is almost at the Lifshitz energy near $W1$ Weyl point, so is $W1$ not contributing to the non-saturation of the magnetization? About analyzing the dHvA data, I hope the authors can provide more details about the carrier density for each pocket and discuss whether they contribute to the torque data in the quantum limit.*

[Reply 2-3]: We thank the reviewer for pointing out this valuable question. In this revised version, we have updated our band structure sketch (Fig. S8) and provided the carrier density for each pocket based on the dHvA data.

The Fermi level is located 27 meV above the W1 point, it is close to the Lifshitz point (about 32 meV above W1) from the band sketch. The electron pockets from W1 with different chirality are well separated in the FS mapping as shown in Fig. S8B, so the chirality of each pocket is well-defined. Therefore, we think that the W1 pocket is also one of the contributors to the non-saturating magnetization, like W2. We also have provided an updated version of our calculations based on the realistic band structure of TaAs (SI section III-E), and added discussions whether and how it contributes to the torque data in the quantum limit on Page 5 in the main text.

Assuming the W1 pocket as an ellipsoid and the W2 pocket as a sphere, we can calculate the carrier density per pocket $n(W1) = 3.96 \times 10^{17} \text{ cm}^{-3}$ and $n(W2) = 3.82 \times 10^{16} \text{ cm}^{-3}$. See section III of the updated SI for more details.

[Remark 2-4]: *For the torque data below the quantum limit, Why is the trend bending up (positive) for TaAs as this is a Weyl semimetal?*

[Reply 2-4]: The trend bending of the torque data in the quantum limit is claimed in Ref. [38] as a signature for Weyl fermions. In [Reply 2-2], we have shown that this criterion is not universal, because it does not exist in other family members of NbAs, such as in NbP, TaP, and TaAs.

Our data of TaP is also similar to the data by Belicas et al in arXiv:1705.00920. We also have added new discussions on the data reliability based on our capacitance torque measurements in a constant-field facility and some other probes in a pulsed-field facility (page 2 in SI, the updated SI section I-B for more details).

[Remark 2-5]: *For the transverse magnetic susceptibility, the author gave a quantum value in Fig. 3B, is there any significance about the number? It was not discussed in the paper. Also, at zero field, is the transverse magnetic susceptibility also very weakly dependent on temperature from 1.5 K to 40 K. I suspect that the temperature dependence is also weak as this is a paramagnetic metal.*

[Reply 2-5]: We thank the reviewer for pointing out this confusing place. Actually, Fig. 3B is not intended to show a quantum value. It is just used to show the linearity of the magnetization data in the quantum limit. A constant differential effective susceptibility indicates the linearity of the magnetization data. Maybe we can say this value is a large number comparing to what observed in Bi Ref. 34. But our analyses revealed

that this value is mainly determined by the particular band structure. We have added related discussion in the main text on page 6. The transverse magnetic susceptibility is also very weakly dependent on temperature. As shown in Fig. 2C and 2D, the low-field M and τ are collapsed together which shows that the transversal magnetic susceptibility are weakly temperature-dependent.

Reply to Reviewer #3

[Remark 3-1]: *The main result of this paper is the observation of a non-saturating torque magnetization at high magnetic field in TaAs, a semimetal known to present Weyl nodes in its band dispersion. The authors conclude that they have found a new signature of 3D massless Weyl fermions in the quantum limit, namely non-saturating linear magnetization. My main objection to this conclusion is that it ignores what is experimentally known about other semimetals, which do NOT host Weyl nodes. The most obvious case is graphite. It is a semimetal with a topologically trivial band structure. When pushed to the quantum limit, it undergoes a field-induced phase transition documented extensively during the last three decades. There is no Weyl fermion in graphite. Yet, its magnetization is featureless, quasi-linear and non-saturating up to 30 T. (See Fig. 4 in Uji et al., Physica B 246-247, 299 (1998)). This looks like very similar to what the present authors find in the case of TaAs. First of all, even in presence of topologically-trivial electrons, one can observe a non-saturating magnetization.*

[Reply 3-1]: We thank the reviewer for this important comment, which helps us to refine our claim and to improve our manuscript. As explicitly shown in Fig. 1 of the main text, for massive fermions, both the parallel and effective transverse magnetization saturate in the quantum limit. In this revised version, we have also calculated the magnetizations for massive fermions even when the gap between the conduction and valence bands is very small. We found that the magnetizations are also saturating in the quantum limit. While for massless fermions without or with a small gap (much smaller than the cutoff energy), they do not saturate. Of course, for massless fermions with a very large gap, they tend to saturate. Therefore, we now conclude that **the relativistic (massless) feature of carriers leads to the non-saturating magnetizations**. This claim should be irrelevant to topologically-trivial or topologically-nontrivial. In this update version, we have rewritten the sentences with the word “topologically-nontrivial” or “topological-trivial” to avoid misunderstanding.

We want to point out that the experimental data of graphite does not contradict with this new claim (also consistent with our original claim). There are band-contact lines in graphite (see Fig. R3 below, adapted from PRB 73, 235112 (2006)), which also gives rise to non-trivial Berry phases. Though these contact lines are accidental, not protected by symmetry, the dispersion near the band-degeneracy points is still massless. It is demonstrated by ARPES that there are relativistic Dirac fermions with

linear dispersion near the Brillouin zone corner H (Nature Physics 2, 595 (2006)), as shown in Fig. R3 (c).

[Redacted]

Figure R3. (a) Sketch of the Fermi surface (half of it) and of the band-contact lines in graphite. A part of the Fermi surface where the electron and hole majorities touch is presented in an enlarged scale on the right; the band-contact lines pass through the conical features of the Fermi surface. (b) The Brillouin zone of graphite is shown. (c) ARPES intensity map taken near the H point. Panels (a) and (b) are adapted from PRB 73, 235112 (2006); Panel (c) is adapted from Nature Physics 2, 595 (2006).

We should emphasize our claim that the non-saturating magnetization in the quantum limit is the signature of the relativistic (massless) feature of materials, irrespective of topologically trivial or nontrivial. We have shown that for massive fermions with parabolic dispersion, the magnetization still saturates, even though the gap between conduction and valence bands vanishes. The non-saturating magnetization found in graphite actually verifies our claim further. Although there are also massive quasiparticles with parabolic dispersion in graphite, the summation of massive and massless contributions also gives the non-saturating feature in the quantum limit. Hence, we believe that the non-saturating magnetization in the quantum limit is a strong signature of the existence of relativistic (massless) fermions and can be applied for both topologically trivial and nontrivial materials.

Conclusion:

We have refined our claim in this new version as “**the non-saturating magnetization in the quantum limit provides a criterion for discerning relativistic fermions (Weyl case or Dirac case with a small gap) and nonrelativistic fermions**”

(parabolic bands with a gap and two parabolic bands touching at a point)". There are relativistic (massless) fermions in graphite with linear dispersion. The non-saturating magnetization found in graphite agrees with our claim.

[Remark 3-2]: *Second, diamagnetism in semimetals (including TaAs) is caused by orbital currents, which may completely overshadow the contribution of Landau-level electrons, the principal player in the theoretical scenario put forward in this paper. How to explain otherwise the absence of any signature in magnetization by an electronic reorganization such an excitonic phase transition (Sci. Rep. 7: 1733 (2017)) in graphite?*

[Reply 3-2]: Sci. Rep. 7, 1733 (2017) shows the observation of an ordered excitonic phase nucleating around the opening of a band gap in graphite. The authors have provided a substantial electrical transport evidence on this excitonic phase. All these data are consistent with each other to support their claim.

For classical electrons, the diamagnetism comes from both valence electrons and free electrons. The orbital currents of valence electrons around atoms can give the diamagnetism. Meanwhile, free electrons coupled to a magnetic field also result in the Landau diamagnetism (for example, see Solid State Physics by N. W. Ashcroft and D. N. Mermin). Then the total magnetization of classical electrons is diamagnetic. For Weyl semimetals with linear dispersion, the low-field susceptibility induced by free electrons is also negative (PRB 94, 195123), i.e., diamagnetism. Because of the formation of the Landau bands, the magnetization M_{\parallel} oscillates with the magnetic field. The oscillation shown in Fig. 4A and the good fitting between the data and our theory indicate that the main contribution comes from the Landau diamagnetism. Furthermore, we mainly focus on the transverse magnetization, which originates from the anisotropy of the energy dispersion and is not relevant to the orbital currents. Our theory applies at extremely strong magnetic fields, where the Landau quantization is a reasonable starting point of description.

Our theory is about one-particle physics. How it can be generalized to many-body effects, such as to excitons, is still not clear to us and out of the scope of this work. We have added a discussion in the revised version as "How our theory of one-particle physics can be applied to many-body effects, such as to excitons [Sci. Re. 7, 1733 (2017)] will be a topic of great interest."

Reviewers' comments:

Reviewer #1 (Remarks to the Author):

The authors have answered my comments in a satisfactory manner and improved the manuscript sufficiently, it may be published in present form.

Reviewer #2 (Remarks to the Author):

In the first version, the author claimed that non-saturating magnetization is the signature of Weyl fermions. However, in the 2nd version, the authors backed up by one step by claiming that it is not just realized in Weyl fermions, but applies to massless fermions in general either in 2D or 3D as it was also observed in graphite. In 3D, it does not distinguish from Weyl semimetal and Dirac semimetal. Therefore, non-saturating magnetization is not a very powerful tool to find Weyl fermion in the future. Not to mention that there are also two trivial pockets that are co-existing with Weyl nodes at the Fermi surface.

The difference between TaAs, NbAs, TaP and NbP that the author put in the reply letter is quite interesting as the claim is quite different from Ref. 38 by Moll et al. For some reasons, the author didn't write it out in the main text or supplementary information. In my opinion, this is very important. I hope the authors could figure out the difference between these four Weyl semimetals. For example, the authors could figure out why only NbAs changes slope in the torque measurement. In my opinion, this will be an important point to be cleared.

To summarize, the current version is still not suitable for the publication in Nature Communication. However, the story might be different if the authors could figure out the difference between four compounds, which will help the community to better understand this family of Weyl semimetals.

Reviewer #3's informal comments:

"... There, I wrote that diamagnetism in semimetals (including TaAs) is caused by orbital currents, which may completely overshadow the contribution of Landau-level electrons, the principal player in the theoretical scenario put forward in this paper. How to explain otherwise that even when there is a phase transition (as in the case of graphite), the magnetic susceptibility (in contrast to all other physical properties) does not show any anomaly?

Apparently, the authors did not grasp the elementary point I was trying to make.

They write: "Our theory is about one-particle physics. How it can be generalized to many-body effects, such as to excitons, is still not clear to us and out of the scope of this work."

But this was not what I had in mind. Let us forget semi-metals! Take an insulator such as silicon or water ice. These are diamagnetic solids (See https://urldefense.proofpoint.com/v2/url?u=https-3A__en.wikipedia.org_wiki_Diamagnetism&d=DwlDAw&c=vh6FgFnduejNhPPDofl_yRaSfZy8CWbWnIf4XJhSqx8&r=bECxf6s9Qd3fOmHsJrDCzP6bIE6k8mfWVKo7VHbK2fY&m=7tLoHPuvbDp3fd7nqSEPX0WmCukUe89FWmN536USH2Y&s=bh09ap19sq9zO4I8F3W4eB4CnjcS1xw3HDRRe4pZIKiQ&e=). If you put them under a strong magnetic field, you will find a linear non-saturating magnetization, which is NOT Landau diamagnetism of mobile electrons.

Nobody would claim that the observation has anything to do with mobile electrons (including Weyl

fermions). Such a behavior would be experimentally indistinguishable from what is seen according to this paper in TaAs (which has roughly 1 electron per 1000 atoms).

...

"

Reply to reviewer 1

Reviewer#1:

'The authors have answered my comments in a satisfactory manner and improved the manuscript sufficiently, it may be published in present form.'

Reply to reviewer #1:

We thank reviewer#1 for the recommendation of publication.

Reply to reviewer 2

Reviewer#2-1:

'In the first version, the author claimed that non-saturating magnetization is the signature of Weyl fermions. However, in the 2nd version, the authors backed up by one step by claiming that it is not just realized in Weyl fermions, but applies to massless fermions in general either in 2D or 3D as it was also observed in graphite. In 3D, it does not distinguish from Weyl semimetal and Dirac semimetal. Therefore, non-saturating magnetization is not a very powerful tool to find Weyl fermion in the future. Not to mention that there are also two trivial pockets that are co-existing with Weyl nodes at the Fermi surface.'

Reply to reviewer #2-1:

We thank the reviewer for these comments. In our first version, we did claim that the non-saturating magnetization can be used for detecting the band crossing, which includes Weyl semimetals. The claims in our original version are not exact because of not including other band cases. With the help of referees' comments, we improve our descriptions and related theoretical treatments in our revised version by including the small gapped case. Our revised criterion is that the non-saturating magnetization can be used for detecting relativistic quasiparticles. Indeed, we did not claim we can distinguish Dirac and Weyl semimetals, even in our first submitted manuscript. For our understandings, the Dirac semimetals will lose its four degeneracy on the Dirac point, turning into Weyl case in the presence of magnetic field.

Because there is no such an existing clear thermodynamic property (the dHvA oscillation is complicated as the SdH oscillations for the determine of Berry phase) for detecting relativistic particles in the Topological semimetal field, we think our paper made a first clear criterion on detecting relativistic particles by a thermodynamic quantity. In addition to the exotic, unexplored physical property of quantum-limit magnetization, we also established an experimental prototype by using the **quantum-limit** torque magneto-metry which is rarely reported in literature. Our discovery will provide a new experimental platform for the Weyl semimetal field, and also attracts more researchers to this exotic quantum-limit probe--torque magnetometry.

Reviewer#2-2:

‘The difference between TaAs, NbAs, TaP and NbP that the author put in the reply letter is quite interesting as the claim is quite different from Ref. 38 by Moll. For some reasons, the author didn't write it out in the main text or supplementary information. In my opinion, this is very important. I hope the authors could figure out the difference between these four Weyl semimetals. For example, the authors could figure out why only NbAs changes slope in the torque measurement. In my opinion, this will be an important point to be cleared. To summarize, the current version is still not suitable for the publication in Nature Communication. However, the story might be different if the authors could figure out the difference between four compounds, which will help the community to better understand this family of Weyl semimetals.’

Reply to reviewer #2-2:

In the revised version, we have added in the Supplementary Information a new section “IV. THE UNIVERSALITY OF THE HIGH-FIELD CRITERION”, to summarize the difference between TaAs family members in the torque and transverse magnetization.

The difference between these four compounds does exist if we compare our results with the claim in Moll’s paper. **However, there is nothing inconsistent for the four compounds in our claim. We have used the extra data for proving the solidarity of our claim.**

We want to reiterate that we focus on a totally different physics from that of Moll’s paper. **The difference pointed by reviewer is clear in Moll’s paper, the**

low-field part is determined by the competing magnetic sources from “Berry para-magnetism” and “Landau diamagnetism”. Then the signal for the low-field side is determined by different samples with different magnitudes of the two competing sources. Here, the “low-field” means “right below the critical magnetic field for entering the quantum limit”. For another physical source of this signal uncertainty for a Weyl/Dirac semimetal, it is demonstrated theoretically with a lattice model that both the magnitude and sign of the orbital contribution of magnetic susceptibility depend on the Fermi energy (see Fig 6 (c) of PRB93, 045201 (2016)). This means that the magnetization of a realistic Weyl semimetal at the weak magnetic field can be either negative or positive depending on the Fermi energy. Therefore, the observation of different signs of magnetizations of these four Weyl compounds, TaAs, NbAs, TaP, and NbP at weak fields relates to their specific material parameters and Fermi energy, but does not relate to the basic energy properties or topology. However, just as we have demonstrated, the behavior of magnetization in the quantum limit is universal irrespective of different material parameters, and thus is more important.

The above-mentioned logic shows that the ‘signal reversal’ is not a universal criterion for Weyl semimetals, this is also accepted by their paper in the discussion part. On the other hand, our paper had provided a consistent result for all the four compounds and formulated a self-consistent and complete theory for the quantum-limit non-saturating magnetization.

Reply to reviewer 3

Reviewer#3:

‘Apparently, the authors did not grasp the elementary point I was trying to make. They write: “Our theory is about one-particle physics. How it can be generalized to many-body effects, such as to excitons, is still not clear to us and out of the scope of this work.” But this was not what I had in mind. Let us forget semi-metals! Take an insulator such as silicon or water ice. These are diamagnetic solids. If you put them under a strong magnetic field, you will find a linear non-saturating magnetization, which is NOT Landau diamagnetism of mobile electrons. Nobody would claim that the observation has anything to do with mobile electrons (including Weyl fermions). Such a behavior would be experimentally indistinguishable from what is seen according to this paper in TaAs (which has roughly 1 electron per 1000 atoms).’

Reply to reviewer #3:

The reviewer considers the core diamagnetization which usually is named as Larmor (Langevin) diamagnetic magnetization. It is well-known that the total magnetization of the non-magnetic solid is the combination of the Pauli paramagnetic magnetization, Landau diamagnetic magnetization, and the Larmor (Langevin) diamagnetic magnetization. However, his/her decision was based on improper statements, in particular the one “orbital currents, which may completely overshadow the contribution of Landau-level electrons” and that on the linear non-saturating magnetization for silicon or water ice. We explain why by the following three points:

1. Our paper aims to distinguish the relativistic and non-relativistic electron in solids, and therefore we do not consider the insulators like silicon or ice as the subjects.
2. In the semimetals in which we are interested, the contribution of the Larmor diamagnetic signal is negligible compared to the contribution of Landau diamagnetism of mobile electrons.

In the textbook by Ashcroft and Mermin, they estimated that the Langevin (Larmor) diamagnetic susceptibility χ_{Langevin} is typically of order 10^{-5} (see the text below Eq. (31.28) of [2]).

[Redacted]

We can also find the data of the large Landau diamagnetization of TaAs in experiment. As shown below, the diamagnetic signal is about 2.4×10^{-4} emu/mol, which is one order of magnitude larger than estimated Larmor diamagnetism. As shown in Fig. 4 A in the main text, the diamagnetic susceptibility in low fields is close to that in high fields, in which the Larmor diamagnetic susceptibility is unchanged. Therefore, we conclude that the contribution of the Larmor diamagnetic signal in TaAs is small, no matter in low

or high fields.

[Redacted]

Fig. 1. The magnetization of TaAs measured in SQUID (from JMMM **408**, 73, (2016))

Actually, it is a common sense that the Landau susceptibility is dominant in semimetals. The reviewer's statement: "Such a behavior would be experimentally indistinguishable from what is seen according to this paper in TaAs (which has roughly 1 electron per 1000 atoms" is not correct. The key point is that the Landau susceptibility is enhanced by the factor m/m^* with m being the free electron mass and m^* being the effective mass (Eq. (31.73) of [2]) in semimetals

$$\frac{\chi_{\text{Landau}}}{\chi_{\text{Pauli}}} \sim \left(\frac{m}{m^*}\right)^2$$

For TaAs, the effective mass is $m^* \sim 0.06m$ (given in the Supplementary Information), therefore $\chi_{\text{Landau}} \gg \chi_{\text{Pauli}} \sim \chi_{\text{Langevin}}$. We can see that in semimetals with less conduction electrons, the Landau diamagnetization is significantly enhanced. This enhancement was also observed in bismuth and graphite.

3. There is no theory about the Larmor diamagnetic torque signal as far as we know. But if we follow the simplest estimation with a classical treatment [1],

$$\chi = -\frac{\mu_0 N Z e^2}{6m} \langle r^2 \rangle.$$

The torque signal should be zero for the spherical closed shell. To say the least, the Larmor diamagnetic torque susceptibility should be independent to the external field. However here we observed a significantly enhanced χ in the quantum limit.

References

- [1] https://urldefense.proofpoint.com/v2/url?u=https-3A_en.wikipedia.org_wiki_Diamagnetism&d=DwIDAw&c=vh6FgFnduejNhPPD0fl_yRaSfZy8CWbWnIf4XJhSqx8&r=bECxf6s9Qd3fOmHsJrDCzP6bIE6k8mfWVKo7VHbK2fY&m=7tLoHPuvbDp3fd7nqSEPX0WmCukUe89FWmN536USH2Y&s=bh09ap19sq9zO4I8F3W4eB4CnjcS1xw3HDRe4pZIKiQ&e=
- [2] N.W. Ashcroft and N. D. Mermin, Solid State Physics (Holt, Rinehart and Winston, New York, 1976).

Reviewers' comments:

Reviewer #1 (Remarks to the Author):

While Reviewer 3 raises serious issues, I find the authors' response to be convincing. I can not say with full confidence who is right in this argument, but I believe it is better in such cases to let the community judge. This work will certainly stimulate further research into magnetic response of Weyl semimetals, which will eventually clarify these and other issues. I recommend this paper for publication.

Reviewer #2 (Remarks to the Author):

In this 2nd revision, the authors provided comparison between the transition metal monopnictide Weyl semimetal family and concluded that they have the same non-saturating magnetization. This behavior also applies to Dirac semimetal and was also observed in graphite. Dirac semimetal is not topological as there is no Chern number at the Dirac point due to the 4-fold degeneracy. Graphite is of course not topological. Therefore, the non-saturating magnetism applies to relativist quasiparticles in general, but not in topological materials. (As a result, the last sentence in the abstract is misleading). The conclusion is basically the same as that of the 2nd version and I still couldn't recommend the publication in Nat. Comm.

Reviewer #4 (Remarks to the Author):

The manuscript by Zhang et al. presents a complete study of the magnetic torque measurements of Weyl semimetal TaAs. Comparing with the experimental results with the theoretical calculation, the authors argued that the non-saturating magnetization serves as a criterion detecting relativistic quasiparticles. Since there are already three full reviews, I will not go over the details but only present my judgment.

Reviewer 3 argued that the core diamagnetism dominates the high field magnetization. This statement is false and irrelevant. The authors present the detailed argument why light mass leads to much larger Landau diamagnetism of electrons. More importantly, the authors measured the magnetic torque that is only sensitive to the magnetic anisotropy. The core diamagnetic contribution is isotropic and thus would barely contribute much to the magnetic torque signal.

Review 2 focused on the definition of topological nature, Weyl vs. Dirac fermions, as well as the difference between NbAs, TaAs, NbP, and TaP. I feel the authors gave good replies and revisions.

My biggest problem with the manuscript is the lack of experimental detail. The regular parallel magnetization is scaled to compare with the calculation. Is there any effort to get the correct magnitude of the parallel magnetization? The unit of magnetic susceptibility χ_T is also wrong. "emu" is a CGS unit for the magnetic moment, not magnetic susceptibility.

I am also dubious about the bold claim of the "a high-field thermodynamic criterion for detecting the magnetic response of relativistic quasiparticles in topological materials." The comparison between the theory and experiments in Fig. 4 is confusing. The model of constant E_F does not explain the quantum oscillations at all. The range of the "Fixed N_c " is hand-picked to look similar to the experimental data, although the starting curvature at 15 T of Panel a is different from the data.

Overall I think the manuscript of Zhang et al. overclaimed the significance of the comparison between their theory and their data. I would not recommend the acceptance of the manuscript.

Reply to reviewer 1

Reviewer#1:

'While Reviewer 3 raises serious issues, I find the authors' response to be convincing. I cannot say with full confidence who is right in this argument, but I believe it is better in such cases to let the community judge. This work will certainly stimulate further research into magnetic response of Weyl semimetals, which will eventually clarify these and other issues. I recommend this paper for publication.'

Reply to reviewer #1:

We thank the reviewer for the recommendation of publication. We believe that our work will have enough impact to the community.

Reply to reviewer 2

Reviewer#2:

'In this 2nd revision, the authors provided comparison between the transition metal monpnictide Weyl semimetal family and concluded that they have the same non-saturating magnetization. This behavior also applies to Dirac semimetal and was also observed in graphite. Dirac semimetal is not topological as there is no Chern number at the Dirac point due to the 4-fold degeneracy. Graphite is of course not topological. Therefore, the non-saturating magnetism applies to relativist quasiparticles in general, but not in topological materials. (As a result, the last sentence in the abstract is misleading). The conclusion is basically the same as that of the 2nd version and I still couldn't recommend the publication in Nat. Comm.

Reply to reviewer #2:

In the previous rounds, we have revised the claim to “relativistic fermions”, not “topological” anymore. Reviewer 4 has seen this difference. According to reviewer 4, “Review 2 focused on the definition of topological nature, Weyl vs. Dirac fermions, as well as the difference between NbAs, TaAs, NbP, and TaP. I feel the authors gave good replies and revisions.” We thank reviewer 2 for spending precious time on our manuscript. We will appreciate if reviewer2 also

acknowledges the revised claim.

Reply to reviewer 4

Reviewer#4-1:

‘Reviewer 3 argued that the core diamagnetism dominates the high field magnetization. This statement is false and irrelevant. The authors present the detailed argument why light mass leads to much larger Landau diamagnetism of electrons. More importantly, the authors measured the magnetic torque that is only sensitive to the magnetic anisotropy. The core diamagnetic contribution is isotropic and thus would barely contribute much to the magnetic torque signal. Review 2 focused on the definition of topological nature, Weyl vs. Dirac fermions, as well as the difference between NbAs, TaAs, NbP, and TaP. I feel the authors gave good replies and revisions.’

Reply to reviewer #4-1:

We thank the reviewer for the objective comments which have brought the review process back to track.

Reviewer#4-2:

‘My biggest problem with the manuscript is the lack of experimental detail. The regular parallel magnetization is scaled to compare with the calculation. Is there any effort to get the correct magnitude of the parallel magnetization? The unit of magnetic susceptibility χ_T is also wrong. "emu" is a CGS unit for the magnetic moment, not magnetic susceptibility.’

Reply to reviewer #4-2:

We thank reviewer 4 for pointing out the misuse of magnetic units.

In the present revised manuscript, we have added the low-field magnetization data carried out by SQUID measurements, as shown in Fig. S10 in SI. Based on the SQUID data, we can rescale the magnetization data obtained in pulsed fields to the absolute unit, as shown in Fig. S10 in SI. The unit of χ_T is also corrected in this version. The units of parallel magnetization and torque-based transverse magnetization are different, which is just based on the traditions of both fields.

Reviewer#4-3:

I am also dubious about the bold claim of the "a high-field thermodynamic criterion for detecting the magnetic response of relativistic quasiparticles in topological materials." The comparison between the theory and experiments in Fig. 4 is confusing. The model of constant E_F does not explain the quantum oscillations at all. The range of the "Fixed Nc" is hand-picked to look similar to the experimental data, although the starting curvature at 15 T of Panel a is different from the data. Overall I think the manuscript of Zhang et al. overclaimed the significance of the comparison between their theory and their data.'

Reply to reviewer #4-3:

In the previous version, we focused on the quantum limit, so the theoretical treatment on the oscillation part was over-simplified. To improve the simulation to well cover both the oscillation and quantum-limit parts, in this revised version, the oscillation part contributed by the conduction bands is included and modeled by a sine function. Together with the non-saturating $B \ln \Gamma$ behavior from the valence bands, we can directly fit the raw data for the whole range of magnetic field with the formula [Eqs. (E93) and (E94) in SI]

$$M_{\parallel} \simeq -\frac{\sqrt{\Delta} \Delta E_F}{8\pi^3 \ell_B^3 B} \sin\left(2\pi \frac{E_F^2 \ell_B^2}{2\Delta v_b^2} - \frac{\pi}{4}\right) \exp\left(-\frac{\Gamma}{B}\right) - \frac{v_b \Delta^2}{24\pi^2 \ell_B^4 B} \left[\ln\left(\frac{2\Lambda}{v_b \sqrt{2\Delta e B}}\right) + \mathcal{A} - \frac{1}{4} \right],$$

$$M_T \simeq \left[-\frac{E_F^2}{8\pi^2 v_b \ell_B^2 B} - \frac{\sqrt{\Delta} E_F}{8\pi^3 \ell_B^3 B} \sin\left(2\pi \frac{E_F^2 \ell_B^2}{2\Delta v_b^2} - \frac{\pi}{4}\right) \exp\left(-\frac{\Gamma}{B}\right) \right] \frac{\partial \Delta(\theta)}{\partial \theta}$$

$$- \frac{v_b \Delta}{24\pi^2 \ell_B^4 B} \left[\ln\left(\frac{2\Lambda}{v_b \sqrt{2\Delta e B}}\right) + \mathcal{A} - \frac{1}{4} \right] \frac{\partial \Delta(\theta)}{\partial \theta}.$$

Further, for the fixed-Nc case, first we use the following equation [Eq. (E34) in SI] to obtain the Fermi energy

$$N_c = \frac{\Delta}{2\pi \ell_B^2} \left[\sum_{n=1, k_{\parallel}} 2f(E_{n+}) + \sum_{k_{\parallel}} f(E_0) \right],$$

then the above fitting equations for M_{\parallel} and M_T can be applied. With the improved simulation, both the low-field oscillations and the quantum limit behaviors of M_{\parallel} are well fitted (The left panel of Fig. R1 below, which is Fig. 4A). The data of M_T is also fitted in the acceptable range, especially for near and in the quantum limit (the right panel of Fig. R1, which is Fig. 4B).

Figure R1. Comparison between the experiment and theory for $M_{||}$ and M_T . The black solid lines are for the experimental data, and the blue and red dash lines are the theoretical simulations at fixed E_F and N_c , respectively.

Now, because the N_c -fixed simulation covers the whole range of magnetic field and E_F is determined at finite temperatures, we do not need to hand pick the range for the “fixed N_c ” simulation.

Despite the improved comparison between the theory and experiment, we have weakened the claim. Specifically, we have changed “criterion” to “method”, and deleted strong claims such as “universal criterion”.

REVIEWERS' COMMENTS:

Reviewer #4 (Remarks to the Author):

The revision addressed all my concerns. I am happy that my review helped in revision.

Reviewer #4 (Remarks to the Author):

The revision addressed all my concerns. I am happy that my review helped in revision.

Authors' reply: Thank the reviewer #4 for recommendation of publication.